# Regional Disparities in Caries Experience and Associating Factors of Ghanaian Children Aged 3 to 13 Years in Urban Accra and Rural Kpando

**DOI:** 10.3390/ijerph19095771

**Published:** 2022-05-09

**Authors:** Anna Peters, Karolin Brandt, Andreas Wienke, Hans-Günter Schaller

**Affiliations:** 1University Outpatient Clinic of Conservative/Restorative Dentistry and Periodontology, Martin-Luther-University Halle-Wittenberg, Magdeburger Str. 16, 06120 Halle (Saale), Germany; karolin.brandt@uk-halle.de (K.B.); hans-guenter.schaller@uk-halle.de (H.-G.S.); 2Institute of Medical Epidemiology, Biostatistics and Computer Science, Martin-Luther-University Halle-Wittenberg, Magdeburger Str. 8, 06120 Halle (Saale), Germany; andreas.wienke@uk-halle.de

**Keywords:** caries prevalence, caries experience, dmft/DMFT, Ghana, significant caries index, significant affected caries Index, social inequality in dental medicine, regional oral health analyses, caries epidemiology, associating factors of dental health

## Abstract

(1) The objective of this socio-epidemiologic cross-sectional study was to investigate caries burdens in Ghanaian children aged 3 to 13 years. The main focus was the analysis of urban–rural disparities and associating socio-demographic and behavioural factors. (2) Standardized caries examination with documentation of decayed, missing, filled deciduous (dmft) and permanent teeth (DMFT) was conducted in 11 school facilities according to WHO guidelines. A parental questionnaire gathered data considering associating factors. Descriptive statistics were used to evaluate their influence on caries prevalence and experience using mean dmft+DMFT, Significant Caries Index (SiC), and Specific Affected Caries Index (SaC). (3) In total, 313 study participants were included (mean age 7.7 ± 3.8 years; 156 urban, 157 rural). The urban region showed slightly higher caries prevalence (40.4% vs. 38.9%). The rural region had higher caries experience in mean dmft+DMFT (1.22 ± 2.26 vs. 0.96 ± 1.58), SiC (3.52 ± 2.73 vs. 2.65 ± 1.71), and SaC (3.15 ± 2.68 vs. 2.37 ± 1.68). Lower education and occupation level of parents and rural residence were associated to higher caries values. Sugary diet showed an inverse relation with caries prevalence and oral hygiene practices supported the generally known etiologic correlation. (4) This study highlights the importance of targeting children vulnerable to caries due to social inequality with adequate preventive means. The implementation of regular dental screening and education, e.g. in schools, may be helpful.

## 1. Introduction

Caries is still one of the most widespread chronic diseases in the world [1,2,3,4]. Due to social inequalities, economic reasons, and differences in healthcare systems of different countries, oral health is not equally accessible to every human in our global society [5]. As described by several institutions, and most recently by the FDI World Dental Federation, the importance of achieving equal conditions for the assessment of healthcare services has increased [6]. In many industrial countries, a strong caries decline has been observed in the past two decades [7,8,9], leading to the development of polarized distributions, which showed that socially disadvantaged families especially experience higher caries burdens [10,11,12]. Particularly in low- and middle-income countries, social and economic inequalities are known to be the reason for the persistently high prevalence in preventable oral diseases [1,13]. In Ghana, where caries experience was low according to past analyses [14,15,16,17,18,19], it is of interest to examine present tendencies of dental health in children as well as the associating factors. Even though dental healthcare has become part of national health insurances in Ghana [20], regular visits to the dentist are uncommon [16,21] especially in societies of low income. A curative, government-oriented, and hospital-based dental care system has been observed in the past rather than shifting preferences towards preventive means and frequent dental examinations [16,22]. Additionally, an urban–rural maldistribution and the long-observed trend towards emigration of trained dentists to industrialized nations due to economic aspects are reasons for the large oral health supply gap in the country [23]. This results in rural regions being even more exposed to effects of less access to proper dental healthcare [23]. At the same time, traditional dietary forms and lower intake of sugary foods and drinks in less industrialized regions are factors that may relativize the effects based on ineffective oral health care [24]. So far, only few studies evaluating the caries experience in children by considering regional, socio-demographic, and behavioural aspects [14,22,24,25,26] have been conducted in Ghana. However, none considered all the previously mentioned aspects at once, which explains the need for investigating present outcomes of these factors in a paediatric population.

This study aimed to compare the caries experience of Ghanaian children aged 3 to 13 years in an urban and rural study population and to consider the effects of different associated factors on caries values. As disease-associated factors, behavioural aspects that are related to the aetiology of caries such as diet and oral hygiene, as well as socio-demographic factors, were considered.

## 2. Materials and Methods

The present study was approved by competent ethics committees in Germany (Research Ethics Committee of the Medical Faculty at the Martin-Luther-University Halle-Wittenberg; reference no.: 2016-149, approved 12 January 2017) and Ghana (Institutional Review Board of 37 Military Hospital Accra; reference no.: 37MH-IRB IPN 119/2017, approved 23 February 2017) prior to its implementation in February and March of 2017. Together with local school authorities (Ghana Education Service), 11 schools and kindergartens (6 government, 5 private) were selected to participate in the study. These were located in the urban region of Accra (country’s capital city in Greater Accra Region) and the rural region of Kpando (Volta Region), forming the two study areas in Ghana. To represent an adequate and widespread image of dental health and to enable concrete differentiation in the age group analyses, the study was conceptualized to include participants of the 3 dentition phases only (primary, mixed, and permanent dentition phase). Therefore, age ranges of 3 to 4 (age group 1), 6 to 7 (age group 2), and 12 to 13 (age group 3) years were applied. After the children were stratified by the required age and randomly selected for participation, detailed study information material was handed out to their parents. The study information sheet explained the content and aim of the study, the voluntary participation and disclosure of personal data, assured the confidential treatment of all information obtained, and included a parental declaration of consent. The informed written consent of a parent was a prerequisite for inclusion in the study. Study participants underwent a dental examination in kindergarten and school facilities after adequately (age-appropriate language) being informed about the procedure to assure the children’s assent. Following the dental examination, an oral health education lesson with practical instructions was given to the entire class. Additionally, an information letter with the result of the dental examination, emphasizing when dental care was needed, was provided to the parents.

Dental examination was carried out in broad daylight, using sterile disposable dental examination instruments (mirror and blunt probe), and under strict hygienic regularities in order to minimize cross-contaminations between study participants. They were conducted by one study examiner (first author) and a dental assistant for documentation, following the WHO guidelines for oral health examinations [27]. During clinical examination, the total amount of decayed (d), missing/extracted due to caries (m), filled (f), deciduous (t), and permanent teeth (T) were documented (total amount of these equals to dmft or DMFT). Caries lesions were detected on the level of cavitation (D3-lesion) [27]. Initial carious lesions (D1 and D2) were not recorded as carious because they are reversible, since they have the possibility of remineralization.

Indices used to depict the occurrence of caries were: caries prevalence (percentual proportion of dmft/DMFT > 0), caries experience (mean dmft/DMFT), Significant Caries Index [28] (SiC = defined as mean dmft/DMFT of one-third of the population with the highest dmft/DMFT-Scores), and Specific Affected Caries Index [29] (SaC = defined as mean value of those with dmft/DMFT > 0). These caries indices were used to describe the caries burden of the subgroups, also dependent on the considered associating factors.

For data collection on these associating factors, an oral hygiene and a dietary parental questionnaire, both to reflect behavioural aspects, as well as the inquiry of socio-demographic aspects (provision of information was voluntary) when obtaining the written consent, were used.

Statistical analysis was conducted to access differences in region (urban/rural), age group (3–4 years, 6–7 years, and 12–13 years), and socio-demographic factors such as family status (single/married), a parent’s educational level (none/low/middle/high), a parent’s occupation (labourer/employee/higher position), and the type of school facility attended by the child (government/private). Parental level of education was categorized as follows: none = illiterate parent, school education non-existent; low = primary school education and junior secondary school education; middle = senior secondary school education; and high = A-level degree and/or academic education (university degree). When parents declared working in unskilled and mainly physically defined jobs (for instance farmers, market personnel) they were considered labourers. The occupation category employee referred to parents working in a profession or an employment relationship that required vocational training, whilst job titles that indicated advanced responsibilities and functions at their workplace combined with academic training (university degree) were ascribed to higher occupational positions.

The oral hygiene questionnaire aimed at assessing the child’s oral hygiene practices. This included when and how often teeth were brushed, whether with parental help or independently, and what instruments and products were used for oral hygiene. The results were categorized into 3 groups: poor (tooth brushing less than 2 times daily), good (tooth brushing at least 2 times daily with fluoridated toothpaste), and very good oral hygiene (tooth brushing at least 2 times daily with fluoridated toothpaste and interdental hygiene using dental floss).

The dietary questionnaire surveyed various components of the child’s nutrition such as meals (breakfast and meals throughout the day), drinks, as well as snacks, and the frequency of their consumption. The sugar content of these components was the primary parameter of the applied evaluation system. To summarize the aspects covered by the questionnaire, an aggregated sugar point system was used that scored all the dietary components surveyed. Therefore, the components (meal, drinks, snacks) were evaluated according to their sugar content and thus classified as cariogenic (=1 sugar point) or non-cariogenic (=0 sugar points). The sum of all evaluations led to a total amount of sugar points, which enabled a dichotomous categorization into low-sugar diet (sum of 0 to 3 sugar points) or high-sugar diet (4 to 6 sugar points) of each study participant.

### Statistical Analysis

Generally, descriptive analysis methods were applied. Data were evaluated using frequency analyses with regard to all the above-mentioned variables of the study population (region, age group, socio-demographic parameters, behavioural factors oral hygiene, and diet) and were performed by cross-tabulation. Consequently, absolute and relative frequencies were reported. Outcomes in caries experience were determined as mean ± standard deviation (SD) for descriptive purposes. For comparison of caries prevalence within the categorical/nominal variables, Chi-Square-test was applied. Fisher’s exact test was used when the expected frequency was ≤5 in a cell of four-field/multi-field tables. For comparison of dmft/DMFT within nominal variables of 2 categories (as region: urban or rural), Mann–Whitney U test was used, and in the case of >2 categories (as age groups 1–3), Kruskal–Wallis test was used. Caries analysis depending on age required using only dmft for the age groups 1 and 2 and only DMFT for the age group 3. For caries analysis, depending on region and the associating socio-demographic and behavioural factors, the total caries burden in dmft+DMFT (both dentitions) was used.

## 3. Results

Out of the 401 initially recruited participants, 313 were included in the evaluations, resulting in a response rate of 87.0%. Participants fell out due to discrepancies in age, leading to them not meeting the guidelines set for this study; missing informed consent of a parent; or simply noncompliance towards dental examination. Study participants were only 3, 4, 6, 7, 12, and 13 years of age with an average age of 7.7 ± 3.8 years. Overall, slightly more study participants were female (*n* = 160, 51.1%) than male (*n* = 153, 48.9%), whilst gender distribution only showed slight differences in the two study areas (Table 1). The socio-demographic determinants displayed strong regional differences when looking at parent’s educational level and occupation. High educational level was most popular in the urban region of Accra (high = 71.9%), whereas the lower levels of education were more common in the rural area of Kpando (middle = 51.6%, low = 14.5%, no education = 4.8%). Similar results were found when looking at the parent‘s occupation: labourers were more common in the rural area (66.0%) and occupations as employees (63.2%) or in higher positions (10.3%) were more common in the urban area, although the proportion of the last mentioned was comparatively low in the overall study population (5.2%). Family status did not underlie great regional differences whereby, in the overall study population, 84.8% of the families were married households, whereas 15.2% were single parents. In total, 55.3% (*n* = 173) of the study population attended private school facilities and 44.7% (*n* = 140) visited government schools. A similar ratio could be observed in both regions.

Caries prevalence was slightly higher in urban Accra (40.4%) than rural Kpando (38.9%). Although a lower share of study participants in rural Kpando had a dmft/DMFT > 0, the mean caries experience (1.22 ± 2.26) as well as SiC (3.52 ± 2.73) and SaC (3.15 ± 2.68) were higher than in urban Accra (mean 0.96 ± 1.58, 2.65 ± 1.71 SiC, 2.37 ± 1.68 SaC), leading to the rural region experiencing a higher caries burden. When looking at the different age groups, age group 2 showed the highest caries prevalence of 45.6% (Table 2). The total caries experience of the age groups 1 and 2 was similar on average (1.23 dmft). Age group 3 had a very low DMFT of 0.67 ± 1.22. It could generally be ascertained that the caries burden is not evenly distributed, but rather affects certain study groups to a greater extent. This effect, known as caries polarization, could be observed throughout the study population.

Considering regional differences within the three age groups, outstanding differences could be observed especially in age group 1, where dmft was 0.74 in the urban and 1.72 in the rural region, and SiC was 2.19 in the urban and 5.0 in the rural region (Figure 1). The SiC used to identify the caries risk groups was higher than the SaC in almost all considered groups of region and age, which can be explained by the fact that caries prevalence was mostly higher than the border mark of 33.3%. Since the highest score in SiC was 5.0 and occurred in age group 1 of the rural region of Kpando, this age group and region was the subgroup with the highest caries risk of this study population.

Socio-demographic aspects were used to stratify the population group by family status, parent’s level of education and occupation as well as by the school facility type. Outcomes in caries prevalence and experience showed moderate differences depending on family status and school facility type in both regions. When parents were married, caries experience was higher in urban Accra (43.4%, 1.02 ± 1.62 mean dmft+DMFT) and lower in rural Kpando (37.0%, 1.12 ± 2.24 mean dmft+DMFT) in comparison to the other form of parenting (Accra: 35.3%, 0.82 ± 1.38 mean dmft+DMFT, Kpando: 45.0%, 1.25 ± 1.97 mean dmft+DMFT). Furthermore, as displayed in Table 3, reverse results regarding school facility type were revealed. In private schools of urban Accra (43.5%, 1.02 ± 1.58 mean dmft+DMFT), caries experience was higher than in government schools (36.6%, 0.87 ± 1.58 mean dmft+DMFT), but in the rural region of Kpando the government school type had slightly higher caries scores (39.1%, 1.26 ± 2.50 mean dmft+DMFT). The factors parental education and occupation showed differences between middle and high education in Accra. Middle education (35.3%, 1.29 ± 2.08 mean dmft+DMFT) was associated with higher caries burdens than the high education level (34.8%, 0.80 ± 1.42 mean dmft+DMFT). In addition, in rural Kpando, the middle education level showed a peak in caries experience (46.9%, 1.72 ± 3.09 mean dmft+DMFT). Here, however, parents also stated they had no education, leading to the highest mean caries experience in the region (33,3%, 2.30 ± 4.00 mean dmft+DMFT). The parents’ occupation level showed similar tendencies: employees’ children in both regions revealed increased caries-scores (Accra: 45.5%, 1.16 ± 1.73 mean dmft+DMFT, Kpando: 40.0%, 1.51 ± 2.87 mean dmft+DMFT). Labourers’ children had lower caries experience (Accra: 34.8%, 0.83 ± 1.44 mean dmft+DMFT, Kpando: 40.0%, 1.16 ± 2.10 mean dmft+DMFT), and higher positions were mainly represented in urban Accra and were accompanied with the lowest caries score in relation to parental occupation level (11.1%, 0.11 ± 0.33 mean dmft+DMFT).

For the associating factor oral hygiene, a clearly opposite gradient to the caries values was recognizable. With improved oral hygiene, caries burdens decreased. This was more obvious in rural Kpando than urban Accra. When considering concrete differences in the regions, the data show this characteristic course with the exception that in participants from the urban region with good oral hygiene, caries prevalence and experience (43.5%, 0.96 ± 1.46 mean dmft+DMFT) was slightly higher than those with poor oral hygiene (38.7%, 0.94 ± 1.63 mean dmft+DMFT).

The evaluation of the individual question components of the diet questionnaire enabled a dichotomous classification into a high-sugar or low-sugar dietary form. The results showed that an inverse gradient for the association between diet and caries burden for this study population emerged. High-sugary diet was occasionally associated with lower caries prevalence, this was particularly evident in urban Accra (high-sugary diet 30.2%, low-sugary diet 45.6%). On the other hand, caries prevalence in rural Kpando did not display great differences when looking at sugar content in diet. Regarding caries experience, the rural region showed the aetiologically known association of caries and diet: a high-sugar diet had higher caries values (1.68 ± 3.18 mean dmft+DMFT, 5.00 ± 3.92 SiC, 4.33 ± 3.87 SaC) than a low-sugar diet (1.11 ± 1.98 mean dmft+DMFT, 3.17 ± 2.29 SiC, 2.86 ± 2.25 SaC). The results in urban Accra showed an opposite correlation and therefore followed the inversed gradient that stood out in caries prevalence.

## 4. Discussion

The present study produced up-to-date data on the caries experience of Ghanaian children and adolescents in a study population from urban Accra and rural Kpando. As the results of this study indicate, caries prevalence showed marginal differences in the urban–rural comparison. Overall, the study participants from urban Accra were slightly more affected by caries but experienced lower caries burdens on dmft+DMFT levels. Caries polarization was strongly discernible since differences in mean dmft and SiC were very prominent. Past studies in Ghana show that caries prevalence varied widely over decades, which could naturally be attributed to the study population itself or to their geographic location [24,26]. Today’s state of research consists of different studies from different regions of the country leading to no available national values. Nevertheless, it can be observed that caries experience based on the collection of DMFT-values in the past was consistently in low ranges, below 1 DMFT on average [16,17,30]. The present study is consistent with previous statements that regional differences in caries prevalence and experience between urban and rural settlements still exist. The studies of Bruce et al. [16] and Beni [17] should be pointed out for comparison since they were carried out in the geographic regions of the present investigation. Accordingly, caries prevalence has surged in general, and regional disparities in the considered areas have levelled out since the rural region of Kpando has experienced a strong increment in caries prevalence (12.5% in 2009 [17] to 38.9% in present study).

In 2002 [16], in an Accra study population, the mean dmft+DMFT was 0.79. The present study determined a value of 0.96 ± 1.58 and thus marked a slight increase. Caries experience was higher in the rural region of Kpando, with an average of 1.22 ± 2.26 dmft+DMFT. A comparison to literature data is only possible in regards to permanent dentition, since Beni et al. [17] only recorded DMFT in their examinations. Therefore, values for both dentitions are not available. With 0.49 ± 0.98 DMFT, the present study revealed an increase in caries experience compared to the result of 2009 [17], which was 0.24 ± 0.75 DMFT.

The present study revealed that the caries experience in rural Kpando was higher than in urban Accra when considering caries experience of the primary and secondary dentition combined (dmft+DMFT). Since previous literature studies did not differentiate between dmft and DMFT, but mostly determined DMFT only or the mixed value of both dentitions, the comparison with these show that the results of this study are in line with the knowledge in regional comparisons implying that rural regions were more affected by caries than urban regions. Moreover, this study showed that caries experience in Ghana has slightly increased over the past few years and thus supports reasonable suspicions in literature describing an increment in caries experience in low- and middle-income countries [4,31,32]. According to the literature, almost 80% of African communities are socially disadvantaged [18]. The existence of widespread poverty and underdevelopment are traits that linger in many African countries and are difficult to shake off. In countries that have experienced an upswing as a result of industrialization, the resulting prosperity only reaches the elite, so that rural, less prosperous areas often do not or hardly benefit from this upswing [33]. In recent decades, connections between oral diseases and influencing factors have increasingly been observed, and their extents have been researched in detail. It has been established that factors such as social status, work, and nutrition can influence oral health. In children, it is the parent-related factors that show influence on caries experience and oral health in general [34,35]. The findings of this study showed that for the educational level of a parent, considering all forms of expression (no, low, middle, and high education), no clear gradient was observed in connection to caries prevalence and experience. Nevertheless, clear differences in caries values between no education and high education could be seen. However, various research projects clearly demonstrate the existence of an inverse relationship between caries prevalence and parental level of education or other indicators of socio-economic levels [36,37,38,39,40], but the expectation of decreasing caries burdens with increasing educational level was not completely fulfilled with the results of this survey. Of course, if the number of investigated subjects is too low, as was the case particularly for the groups of no education and low education (n=3 and n=10), this can have a negative effect on sound results and prevent a well-founded interpretation. Due to the low response rate for the voluntary socio-demographic information given by the parents, a reduced readiness to provide this kind of information can be surmised. We assume that information about education, family status, and occupation may be considered as too vulnerable for Ghanaian parents and thus lead to them refusing to give certain information. With only 126 subjects providing information about their educational degree, the analysis was able to reflect just about half of the total number of study participants examined.

Oral hygiene represents a classic risk factor for the development of caries. It determines the extent of the formation of dental plaque and is the decisive regulating factor for the development and progression of carious lesions, especially in a high-sugar diet. Within this knowledge, it has been shown, that not the oral hygiene practices themselves, but the content of fluoride in the agents used (e.g., toothpaste) are responsible for caries-protective effects [41,42]. The described relation between oral hygiene and caries burdens in the results of this study provide a basis for the existence of potential effects of good oral hygiene practices on caries values, but it must be considered that significant evidence was lacking. The possible effects could also be attributed to the fluoride content of the applied toothpaste, rather than to the plain effects of mechanical removal of dental plaque during tooth brushing. Studies investigating adequate oral hygiene show that the frequency of brushing, the cleaning agents used, the time of the day, as well as the time spent brushing are directly related to the development of caries [43]. The attitude and behaviour towards daily cleaning play decisive roles, especially in children and adolescents [44]. According to current studies, establishing good oral hygiene in early childhood significantly improves later practices in adulthood [45]. Cleaning at least twice a day for at least 2 min with a toothbrush and fluoridated toothpaste is considered adequate [46]. The implementation of interdental cleaning has been described as an indicator for high motivation in oral hygiene [47,48]. Good oral hygiene appears to prevent caries only to a certain degree, since the complete removal of plaque, especially in the interdental area, is impossible—even when using dental floss. Within the interaction of diet and oral hygiene, “good” oral hygiene practices may balance the effects of cariogenic diet and therefore limit the formation of caries [49]. However, it should be noted that adequate instruction on oral hygiene to prevent caries is an important aspect but should not solely be focused on in caries-preventive measures.

Another major behavioural factor for the formation of dental caries is diet, whereby the amount and frequency of the consumption of sucrose plays a decisive role [50,51]. The results of this study showed the expected connection that a high-sugar diet is associated with an increased burden of caries in the rural region of Kpando only. The urban region, surprisingly, displayed the opposite association. According to previous studies, diet in Ghana is generally very much influenced by traditional food, resulting in no great differences in the components of meals throughout the regions of the country [22]. Furthermore, a study analysing the acidogenic potential of two typical Ghanaian meals revealed that these had non-acidogenic potential and thus suggested that they may have non-cariogenic or even cariostatic effects [52]. The described characteristic pattern of the past, that rural regions mainly stick to traditional, less sugary foods [16] and thus experience less dental caries, must be relativized since changes in dietary forms have generally been observed in African countries [53]. Urban regions especially have experienced changes in dietary forms with commonly increased access to industrialized foods, which are usually more saccharated [53]. A transition in nutrition has not only been observed in Ghana but also other countries of the West African Region. The consequences experienced are obesity, hypertension, and diabetes [53]. In addition, increasing effects in other non-communicable diseases such as caries seem to be likely and have been suggested in the past, but present tendencies need further investigation [54].

Even though the applied assessment of dietary habits in this study was carried out equally and systematically, it should be noted that investigations without methods applying laboratory-based nutritional analysis can be highly subjective. Since it is well known that studies carrying out dietary analysis can be very complex, it can be concluded that the methodology for recording dietary aspects in this study was not able to adequately depict the issue. An adequate method for analysing the connection between diet and caries in Ghana is therefore rather intricate and demands more precise analytical methods.

Generally, methods for analysing the effects of behavioural aspects such as diet or oral hygiene on caries burdens in at-risk children can be very difficult, and there is need to improve the methodology of designs on this topic [55].

In Germany, the legal basis for serial dental examinations of children has been established since 1989. This group prophylaxis programme covers measures to detect and prevent dental disease in children and adolescents up to 12 or 16 years of age, respectively, and is operated by the country’s health authorities [56]. The actions taken within this programme in educating the population about oral-health-promoting measures, means of fluoridation, as well as the effective dental care system with emphasis on preventive approaches have led to a massive reduction in caries in recent decades, especially among adolescents [57]. It has been shown that Germany and other countries experienced a decline in caries, and all of them took similar prevention actions [58,59,60]. Since health measures of this kind make it possible to reach large sections of the population and may increase access to dental examinations for socially disadvantaged groups, widespread prevention programmes indicate a great potential for the Ghanaian healthcare system. Generally, especially for oral diseases such as caries, simple preventive means and access to regular dental examinations may help reduce the burden. Prevention through a combination of community oral health programmes and professionally instructed individual action has shown positive effects. A combination of different preventive means, seems to be the key aspect for reducing the risk of caries development in low- and middle-income countries [31]. According to an evaluation of school based oral-health projects of the WHO and several other studies carried out to evaluate the matter, schools provide an adequate environment to support the promotion of children’s health [61,62,63]. The survey emphasised that the health promoting potential is not fully being exploited globally and that, especially in low- and middle-income countries, the need is urgent.

### Study Limitations

Since this study is a cross-sectional study, the descriptive nature of this study type needs to be emphasized. Cross-sectional studies enable the examination of disease prevalence and other numerous characteristics at once. The particular variables are gathered at one specific point in time; thus, the variables are not manipulated during the observation period. This may help draw conclusions about healthcare service needs in a population, but it is of importance when interpreting the results that this study type does not enable causal determinations, since it is purely observational. Cross-sectional studies are an important tool in epidemiology to quickly create facts about the extent and spread of a disease and the factors associated with it. The gathered data within the analysis of associating factors can be supportive for further research.

Another limitation was that the study population was a convenience sample, meaning that a randomized selection of participating schools was not carried out. We solely randomly selected the study participants within the participating schools.

In addition, the use of questionnaires is a method that can generally lead to bias since data provided by study participants themselves or by their parents represent third-party data. Generally, the accuracy of self-reported data can unconsciously or consciously be embellished. In questionnaires, socially desirable answers may be given by respondents, which may lead to potentially distorted results.

This study is an investigation carried out in two regions of Ghana (urban region of Accra and rural region of Kpando). Data are therefore only shown for these regions and, considering this limitation, cannot be transferred to the whole country of Ghana. A country-representative examination could not be carried out due to organizational and financial reasons.

## 5. Conclusions

A disease is not only characterized by its physical dimension but also by psychological and social determinants. Constantly examining all these dimensions and their impact on caries formation is necessary to manage the burden in populations. In Ghana, annual school examinations, analysing disease-associated aspects, and conducting oral health education in school classes could facilitate the management of caries burdens in the country. This study aimed at contributing to the establishment of such programmes in Ghana. Regular dental examinations combined with oral health education for children and their parents about the most important dental diseases is an appropriate measure to reduce the spread of these in the Ghanaian population. The demand for the integration of oral disease prevention programmes as part of national public health programmes is undeniable, and its implementation in Ghana is strongly necessary. In addition to the existing urban–rural disparities, the 6–7-year-olds of this study population had the highest caries prevalence and participants between 3 and 4 years old had the highest mean caries experience. Furthermore, families of low education and occupation level and living in a rural area were associated with higher caries values. These groups should therefore be addressed more intensively when carrying out oral health prevention programmes.

## Figures and Tables

**Figure 1 ijerph-19-05771-f001:**
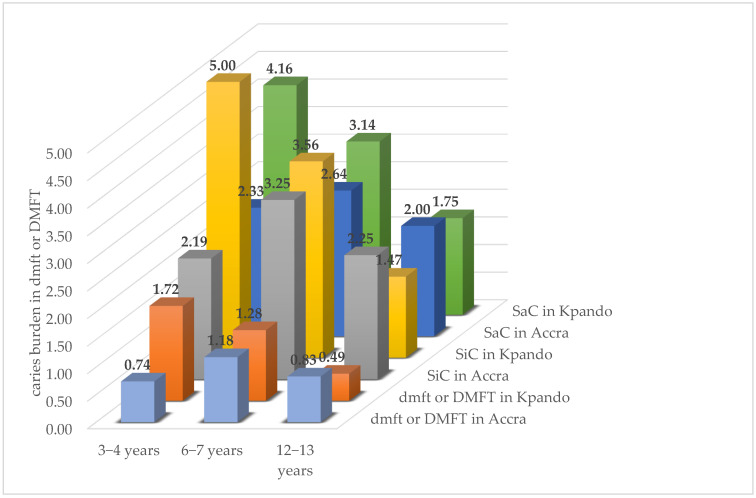
Regional comparison of caries experience (mean dmft in 3–4- and 6–7-year-olds or DMFT in 12–13-year-olds, accordingly, for SiC and SaC) in all three age groups.

**Table 1 ijerph-19-05771-t001:** Socio-demographic determinants of the study population.

Variable (n, *%*) or (Mean ± SD)		Urban Accra *n* = 156		Rural Kpando *n* = 157		*p*-Value	Total Study Population *n* = 313	
age (years)		7.7 ± 3.9		7.7 ± 3.7		-	7.7 ± 3.8	
sex	female	83	53.2	70	44.6	0.127	153	48.9
	male	73	46.8	78	55.4		160	51.1
^1^ family status (69 missing)	single parent	17	14.7	20	15.6	0.833	37	15.2
	married parents	99	85.3	108	84.4		207	84.8
^1^ parent’s education level (187 missing)	none	0	0	3	4.8	<0.001	3	2.4
	low	1	1.6	9	14.5		10	7.9
	middle	17	26.6	32	51.6		49	38.9
	high	46	71.9	18	29.0		64	50.8
^1^ parent’s occupation level (120 missing)	labourer	23	26.4	70	66.0	<0.001	93	48.2
	employee	55	63.2	35	33.0		90	46.6
	higher position	9	10.3	1	0.9		10	5.2
school facility type	state	71	45.5	69	43.9	0.781	140	44.7
	private	85	54.4	88	56.1		173	55.3

^1^ optional information in parental questionnaire.

**Table 2 ijerph-19-05771-t002:** Outcomes in caries prevalence and experience (mean dmft+DMFT, SiC, and SaC) dependent on region and age.

		Caries Prevalence			Caries Experience					
Variable		*n*	*%*	*p*-Value	dmft	DMFT	dmft+DMFT	SiC	SaC	*p*-Value
region	urban Accra	156	40.4	0.782	0.97 ± 1.65	0.83 ± 1.39	0.96 ± 1.58	2.65 ± 1.71	2.37 ± 1.68	0.855
	rural Kpando	157	38.9		1.48 ± 2.56	0.49 ± 0.98	1.22 ± 2.26	3.52 ± 2.73	3.15 ± 2.68	
age	3–4 years	93	36.6	0.313	1.23 ± 2.37	-	-	3.58 ± 2.93	3.35 ± 2.89	0.202
	6–7 years	103	45.6		1.23 ± 1.99	-	-	3.42 ± 2.12	2.86 ± 2.12	
	12–13 years	117	36.8		*-*	0.67 ± 1.22	-	1.95 ± 1.39	1.90 ± 1.37	
total		313	39.6		1.23 ± 2.17	0.67 ± 1.22	1.09 ± 1.95	3.09 ± 2.31	2.75 ± 2.25	

**Table 3 ijerph-19-05771-t003:** Outcomes in caries prevalence and caries experience (mean dmft+DMFT, SiC, and SaC) depending on family status, parent’s educational level, parent’s occupation level, and school facility type attended by the study participant, oral hygiene, and diet.

Variable (*n*, %)		Urban Accra *n* = 156	Effect				Rural Kpando *n* = 157	Effect				*p*-Value	
			Caries Prevalence (%)	dmft+DMFT	SiC	SaC		Caries Prevalence (%)	dmft+DMFT	SiC	SaC	For Caries Prevalence	For dmft+DMFT
^1^ family status	single parent	17	35.3	0.82 ± 1.38	2.33 ± 1.37	2.33 ± 1.37	20	45.0	1.25 ± 1.97	3.29 ± 2.14	2.78 ± 2.11	0.549	0.525
	married parent	99	43.4	1.02 ± 1.62	2.76 ± 1.75	2.35 ± 1.70	108	37.0	1.12 ± 2.24	3.25 ± 2.87	3.03 ± 2.81	0.348	0.527
^1^ parent’s educational level	none	0	-	-	-	-	3	33.3	2.30 ± 4.00	7.00	7.00	-	-
	low	1	0	0	0	0	9	44.4	0.55 ± 0.73	1.33 ± 0.58	1.25 ± 0.50	1.0	0.426
	middle	17	35.3	1.29 ± 2.08	3.7 ± 1.86	3.67 ± 1.86	32	46.9	1.72 ± 3.09	4.63 ± 3.88	3.67 ± 3.68	0.436	0.633
	high	46	34.8	0.80 ± 1.42	2.40 ± 1.55	2.31 ± 1.54	18	16.7	0.50 ± 1.20	1.50 ± 1.76	3.00 ± 1.00	0.154	0.218
^1^ parent’s occupation level	labourer	23	34.8	0.83 ± 1.44	2.38 ± 1.51	2.38 ± 1.51	70	40.0	1.16 ± 2.10	3.30 ± 2.55	2.89 ± 2.47	0.656	0.625
	employee	55	45.5	1.16 ± 1.73	3.17 ± 1.69	2.56 ± 1.73	35	40.0	1.51 ± 2.87	4.25 ± 3.60	3.79 ± 3.51	0.611	0.876
	higher position	9	11.1	0.11 ± 0.33	0.33 ± 0.58	1.0	1	100	2.0	-	2.0	0.200	0.025
school facility type	government	71	36.6	0.87 ± 1.58	2.50 ± 1.82	2.38 ± 1.79	69	39.1	1.26 ± 2.50	3.61 ± 3.23	3.22 ± 3.12	0.759	0.611
	private	85	43.5	1.02 ± 1.58	2.79 ± 1.64	2.35 ± 1.62	88	38.6	1.19 ± 2.08	3.45 ± 2.32	3.09 ± 2.31	0.513	0.828
oral hygiene	poor	62	38.7	0.94 ± 1.63	2.62 ± 1.86	2.42 ± 1.82	93	44.1	1.47 ± 2.53	4.10 ± 2.95	3.34 ± 2.89	0.506	0.309
	good	85	43.5	0.96 ± 1.46	2.61 ± 1.47	2.22 ± 1.46	64	31.3	0.86 ± 1.76	2.62 ± 2.22	2.75 ± 2.20	0.127	0.208
	very good	7	14.3	0.29 ± 0.76	1.0 ± 1.41	2.0	0	-	-	-	-	-	-
diet	high-sugary	53	30.2	0.85 ± 1.70	2.50 ± 2.12	2.81 ± 2.04	31	38.7	1.68 ± 3.18	5.00 ± 3.92	4.33 ± 3.87	0.424	0.333
	low-sugary	103	45.6	1.01 ± 1.51	2.68 ± 1.57	2.21 ± 1.53	126	38.9	1.11 ± 1.98	3.17 ± 2.29	2.86 ± 2.25	0.304	0.603

^1^ optional information in parental questionnaire.

## Data Availability

Data can be provided by the corresponding author upon request.

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
