# Peer review of "Regional Disparities in Caries Experience and Associating Factors of Ghanaian Children Aged 3 to 13 Years in Urban Accra and Rural Kpando"

_ijerph, 2022, doi:10.3390/ijerph19095771_

Round 1

Reviewer 1 Report

This is a cross-sectional study to assess caries burden in Ghanaian children. The knowledge presented is new and may shed some light in a reality that is not well known.

Some issues need to be solved to improve the quality of this study:

Abstract structure needs to be revised and the meaning of sentences in the results part of the abstract is not clear or comprehensible.

Introduction needs to end with the goals of the present study.

Author Response

Response to Reviewer 1

Thank you very much for your review. We have made some amendments to the manuscript according to your recommendations. Our changes have been highlighted in blue colour in the resubmitted manuscript.

Point 1: Abstract structure needs to be revised and the meaning of sentences in the results part of the abstract is not clear or comprehensible.

Response 1: The results part of the abstract has been changed as follows to convey the message more comprehensibly: “Sugary diet showed an inverse relation with caries prevalence and oral hygiene practices supported the generally known etiologic correlation.”

Point 2: Introduction needs to end with the goals of the present study.

Response 2: We have added the goals of the study in the last paragraph of the introduction section: “This study aimed to compare the caries experience of Ghanaian children aged 3 to 13 years in an urban and rural study population and to consider the effects of different associated factors on caries values. As disease-associated factors, behavioural aspects that are related to the etiology of caries such as diet and oral hygiene, as well as socio-demographic factors were considered.” 

Reviewer 2 Report

The manuscript is appropriately written. However, there are few comments in order to improve the manuscript.

Comments on Title, Abstract, and Keywords:

  • The title is suitable.
  • The abstract summarizes well the manuscript.
  • Keywords are suitable.

Comments on Introduction:

  • The introduction is suitable and have no further comments.

Comments on Methods:

  • The methodology used in the manuscript is well explained.

Comments on Results:

  • The results are very systematically and nicely presented.
  • Please present the tables more appropriately, as they are very confusingly displayed.

Comments on Discussion and Conclusions:

  • I would suggest shortening the first paragraph when reporting the main findings, as it was already extensively presented in the results section.
  • Please revise this section grammatically.
  • In the conclusions, could you please explain what were the “dimensions of a disease”?
  • In addition, I suggest deleting lines 397-404 as the information presented is duplicated.

Comments on References:

  • Authors of the manuscript presented relevant references. They have cited a variety of research groups and, therefore, references could be helpful for other researchers. However, references should be carefully revised according to author guidelines.

Author Response

Response to Reviewer 2

Thank you very much for your review. We have made some amendments to the manuscript according to your recommendations. Our changes have been highlighted in blue colour in the resubmitted manuscript.

Point 1: Comments on Results: The results are very systematically and nicely presented. Please present the tables more appropriately, as they are very confusingly displayed.

Response 1: Thank you for your advice. We have changed the formatting of the tables as follows:

  • formatting in all tables: the variables have been formatted to be left-justified and the values have been arranged in the centre.
  • table no. 2 has been formatted to appropriately fit the columns.
  • table no. 3 has been formatted to fit the landscape format of the page for more clarity.

Point 2: Comments on Discussion and Conclusions: I would suggest shortening the first paragraph when reporting the main findings, as it was already extensively presented in the results section.

Response 2: Thank you, we have shortened the first paragraph of the discussion. Thereby we concentrated on the results in regional comparison only since this was the content of the following discussion. Please see the changes we have made in the resubmitted manuscript (they are highlighted in blue colour).

Point 3: Comments on Discussion and Conclusions: Please revise this section grammatically.

Response 3: Thank you, we have revised the discussion and conclusion section grammatically and made some changes to the resubmitted manuscript (they are highlighted in blue colour).

Point 4: Comments on Discussion and Conclusions: In the conclusions, could you please explain what were the “dimensions of a disease”?

Response 4: Thank you for your suggestion. The respective paragraph has been amended as follows: “A disease is not only characterized by its physical dimension, but also by psychological and social determinants. Constantly examining all these dimensions and their impact on caries formation is necessary to manage the burden in populations. In Ghana, annual school examinations, analysing disease-associated aspects, and conducting oral health education in school classes could facilitate the management of caries burdens in the country.”

We hope that changing this paragraph at the beginning of the conclusion of our manuscript conveys our message that research into all these dimensions are of great relevance/importance for understanding and managing the caries disease in populations.

Point 5: Comments on Discussion and Conclusions: In addition, I suggest deleting lines 397-404 as the information presented is duplicated.

Response 5: Thank you very much for your suggestion. After careful revision of this part of the manuscript, we have decided to shorten the respective lines. In the conclusion, we would nevertheless like to highlight which population groups need to be targeted more intensively in caries prevention measures based on our study results, as we believe that the domestic readership could benefit from this.

Point 6: Comments on References: Authors of the manuscript presented relevant references. They have cited a variety of research groups and, therefore, references could be helpful for other researchers. However, references should be carefully revised according to author guidelines.

Response 6: Thank you for your suggestion. We have checked the author guidelines of the Journal with regard to citation style. We used the journal's default citation style and therefore did not need to make any changes.

Reviewer 3 Report

Dear authors,

I've read the manuscript with great interest. It raises a very important issue. In my opinion, it is a well-written paper. However, I see a few things which need to be corrected.

Introduction

The objectives of this study must be highlight

Material and Methods

How it has happened that a German researcher conducted the research in Ghana? Was it a humanitarian mission or another project?

Results

Table 3 needs to be formatted-there are different sizes of the font-please check it.

Please add the limitation of the study

Author Response

Response to Reviewer 3

Thank you very much for your review. We have made some amendments according to your recommendations:

Point 1: Introduction: The objectives of this study must be highlight

Response 1: Thank you for this suggestion. We have added a paragraph to the introduction section considering the objectives of this study: “This study aimed to compare the caries experience of Ghanaian children aged 3 to 13 years in an urban and rural study population and to consider the effects of different associated factors on caries values. As disease-associated factors, behavioural aspects that are related to the etiology of caries such as diet and oral hygiene, as well as socio-demographic factors were considered.” 

Point 2: Material and Methods: How it has happened that a German researcher conducted the research in Ghana? Was it a humanitarian mission or another project?

Response 2: This study was conducted as a research project because the topic was of great interest to the researchers. Since previous studies about caries burdens in Ghanaian children were conducted several years ago (as can be seen in the references), up-to-date data was necessary. In Germany, school-based caries prevention programmes are well-established and it was of interest of the first author to conduct a project like that in Ghana.

Point 3: Results: Table 3 needs to be formatted-there are different sizes of the font-please check it.

Response 3: Thank you for your advice. We have made some changes to table 3: it has been formatted considering font size (all size 10) and general formatting. Additionally, the table now fits the landscape format of the page so that it appears more clearly arranged.

Point 4: Results: Please add the limitation of the study.

Response 4: Thank you for this important advice. We have added a section ‘Study limitations’ in the last paragraph of the discussion:

Since this study is a cross-sectional study, the descriptive nature of this study type needs to be emphasized. Cross-sectional studies enable the examination of disease prevalences and other numerous characteristics at once. The particular variables are gathered at one specific point in time, thus the variables are not being manipulated during observation period. This may help draw conclusions about health care service needs in a population, but it is of importance when interpreting the results that this study type does not enable causal determinations since it is purely observational. Cross-sectional studies are an important tool in epidemiology to quickly create facts about the extent and spread of a disease and the factors associated with it. The gathered data within the analysis of associating factors can be supportive for further research. Another limitation was that the study population was a convenience sample, meaning that a randomized selection of participating schools was not carried out. We solely randomly selected the study participants within the participating schools. In addition, the use of questionnaires is a method that can generally lead to bias since data provided by study participants themselves or by their parents represent third-party data. Generally, the accuracy of self-reported data can unconsciously or consciously be embellished. In questionnaires socially desirable answers may be given by respondents, which may lead to potentially distorted results. This study is an investigation carried out in two regions of Ghana (urban region of Accra and rural region of Kpando). Data is therefore only shown for these regions and, considering this limitation, cannot be transferred to the whole country of Ghana. A country-representative examination could not be carried out due to organizational and financial reasons.

Reviewer 4 Report

Very well done.

One aspect that is missing is the access to water. How is the access to clean water in the rural community ? Oral hygiene practices need to analyzed not just on numbers, since important issues like access to clean water, fluoridated toothpaste and even having individual brushes are fundamental challenges in many communities . 

Author Response

Response to Reviewer 4

Point 1: One aspect that is missing is the access to water. How is the access to clean water in the rural community ? Oral hygiene practices need to analyzed not just on numbers, since important issues like access to clean water, fluoridated toothpaste and even having individual brushes are fundamental challenges in many communities. 

Response 1: Thank you very much for your review as well as the interesting aspects you have brought forward. This paper aimed at accessing socially associating factors as well as behavioural aspects (oral hygiene and diet) on caries burdens of Ghanaian children. Certainly you are right with your objection. Access to clean water as well as fluoride in toothpaste and the possession of appropriate oral hygiene means are aspects that can influence oral health with regard to caries formation. All the aspects covered by the questionnaire and thus all information given by the study parents were used to categorize the study participants regarding their oral hygiene practices in poor, good and very good oral hygiene. The applied oral hygiene questionnaire also covered some of your mentioned aspects substantively. With regard to the aspects you stated, the questionnaire did not reveal differences in the regions. In detail a total of 97.1% of study participants used a manual toothbrush for oral hygiene and 99.4 % stated to use toothpaste. We could not observe great regional differences within these aspects. The access to clean water was not covered by the questionnaire. Differences could only be observed with regard to frequency of toothbrushing per day and the use of dental floss for interdental cleaning. Therefore mainly these characteristics stood out for the categorization of oral hygiene.